# Fusicoccin (FC)-Induced Rapid Growth, Proton Extrusion and Membrane Potential Changes in Maize (*Zea mays* L.) Coleoptile Cells: Comparison to Auxin Responses

**DOI:** 10.3390/ijms22095017

**Published:** 2021-05-09

**Authors:** Małgorzata Polak, Waldemar Karcz

**Affiliations:** Institute of Biology, Biotechnology and Environmental Protection, Faculty of Natural Sciences, University of Silesia in Katowice, Jagiellońska 28, PL-40032 Katowice, Poland; malgorzata.m.polak@us.edu.pl

**Keywords:** Auxin (IAA), coleoptile segments, growth, fusicoccin, membrane potential, proton extrusion

## Abstract

The fungal toxin fusicoccin (FC) induces rapid cell elongation, proton extrusion and plasma membrane hyperpolarization in maize coleoptile cells. Here, these three parameters were simultaneously measured using non-abraded and non-peeled segments with the incubation medium having access to their lumen. The dose–response curve for the FC-induced growth was sigmoidal shaped with the maximum at 10^−6^ M over 10 h. The amplitudes of the rapid growth and proton extrusion were significantly higher for FC than those for indole-3-acetic acid (IAA). The differences between the membrane potential changes that were observed in the presence of FC and IAA relate to the permanent membrane hyperpolarization for FC and transient hyperpolarization for IAA. It was also found that the lag times of the rapid growth, proton extrusion and membrane hyperpolarization were shorter for FC compared to IAA. At 30 °C, the biphasic kinetics of the IAA-induced growth rate could be changed into a monophasic (parabolic) one, which is characteristic for FC-induced rapid growth. It has been suggested that the rates of the initial phase of the FC- and IAA-induced growth involve two common mechanisms that consist of the proton pumps and potassium channels whose contribution to the action of both effectors on the rapid growth is different.

## 1. Introduction

Although research on the effects of auxin (IAA) on plant growth has been conducted for quite some time, the molecular mechanism of auxin action on ion transport in growing cells has still not been precisely explained. According to the so-called “acid growth theory” of auxin action, which was independently proposed by Cleland [1] and Hager et al. [2], auxin activates the plasma membrane (PM) H^+^-ATPase, which acidifies the apoplast and causes the activation of the enzymes that are involved in cell wall loosening (for a review, see [3,4,5]. The theory still evokes discussion and has undergone multiple modifications due to the large amount of new information on the molecular mechanisms of auxin action on plant cell growth [6,7,8].

Patch-clamp techniques that were applied to maize coleoptile protoplasts showed that auxin-induced growth in maize coleoptile cells involves K^+^ uptake via voltage-dependent, inwardly rectifying K^+^ channels (ZMK1, Zea mays K^+^ channel 1), the activity of which contributes to water uptake and consequently to cell expansion (reviewed in [5,9]). The ZMK1 channels exhibit the typical properties of voltage-dependent, proton-stimulated K^+^ channels, which are activated by a hyperpolarizing membrane potential and by extracellular apoplastic protons [9,10,11]. These channels mediate the K^+^ uptake into the cortex cells, thereby increasing their turgor and cell expansion. It has been shown that, apart from the posttranslational, auxin-dependent upregulation of the K^+^ uptake channels, auxin also regulates the expression of the maize K^+^ uptake channel gene *ZMK1* [10]. In turn, this leads to an increase in the number of active K^+^ channels in the plasma membrane [10]. Interestingly, a scenario similar to that of the auxin-induced activation of the K^+^ channels was reported earlier to stimulate H^+^-pumping ATPase in the plasma membrane of maize coleoptile cells [12]. Experiments that were performed using the patch-clamp technique confirmed earlier studies that showed that auxin-induced growth strictly depends on the external K^+^ supply [13]. The idea that the K^+^ uptake channels are crucial for auxin-induced growth also comes from experiments in which cell elongation and K^+^ conductance appeared to be sensitive to extracellular calcium (Ca^2+^) and tetraethylammonium (TEA), which is a K^+^ channel blocker [14,15].

The phosphorylation of the penultimate amino acid Thr in the C terminus of the H^+^-ATPase and the subsequent binding of a 14–3−3 protein to the phosphorylated C terminus is the major common mechanism by which H^+^-ATPase is activated in plant cells [16,17]. It should also be noted that H^+^-ATPase is phosphorylated at multiple sites besides the penultimate Thr [17,18]. In a recent study, Takahashi et al. [19] demonstrated that the application of the natural auxin indole-3-acetic acid (IAA) to endogenous auxin-depleted hypocotyl segments of *Arabidopsis thaliana* induced the phosphorylation of the penultimate threonine of H^+^-ATPase and increased its activity without altering the amount of the enzyme. Changes in both the phosphorylation level of H^+^-ATPase and IAA-induced elongation were similarly concentration dependent. The findings obtained by Takahashi et al. [19] define the activation mechanism of H^+^-ATPase by auxin during the early-phase hypocotyl elongation and, according to the authors, this is the long-sought-after mechanism that is central to the acid-growth theory.

The phytotoxin fusicoccin (FC), which is a diterpene glycoside that is produced by the fungus *Fusicoccin amygdali* Del. [20], mimics the effects of auxin in many respects [21], although its mode of action at the molecular level differs from that of auxin. It has been well documented that FC binds to the H^+^-ATPase/14−3−3 complex and stabilizes it, thus causing an increase in the H^+^ pump activity [22,23,24,25]. It has recently been shown that the K^+^ inward rectifier KAT1 (K^+^ *Arabidopsis thaliana* 1) channel is regulated by the 14−3−3 proteins and that it is further modulated by fusicoccin (FC) [26]. Using patch-clamp technique, these authors performed experiments in whole-cell and inside-out configurations, which showed that 14−3−3 binding augments the KAT1 conductance by increasing the maximal current and by positively shifting the voltage dependency of gating. Fusicoccin potentiated the 14−3−3 effect on the KAT1 activity by stabilizing their interactions [26]. The authors suggested that the 14−3−3 proteins might function as pivotal regulators of ion transport, thereby integrating different stimuli in generating and maintaining the plasma membrane potential.

The patch-clamp experiments that were recently performed by Burdach et al. [27,28] should also be mentioned. These authors showed that the SV (slow vacuolar) and FV (fast vacuolar) channels, which represent the conductance of the major cations across the tonoplast, are involved in the IAA-induced volume changes of the vacuoles.

To understand how auxin signals, particularly the ion transport-dependent ones, are transduced at a cellular level to elicit growth responses, we compared the results of the experiments that investigated the effects of FC and IAA on the growth, medium pH and membrane potential of maize coleoptile cells. These three parameters are fundamental for the “acid growth theory” of auxin action [5] and, here, were measured simultaneously on the same tissue samples. The results presented here show that the rate of the initial phase of the FC- and IAA-induced growth involves two common mechanisms that consist of the proton pumps and potassium channels whose contribution to the action of both effectors on the rapid growth is different.

## 2. Results

### 2.1. FC-Induced Elongation Growth of the Maize Coleoptile Segments and Medium pH Changes Measured Simultaneously with the Growth

High-resolution measurements of the growth were performed using an apparatus that was recently described by Polak and Karcz [8]. Simultaneously with growth medium pH changes were also measured. Figure 1A shows the growth-promoting activity of 10^−7^, 10^−6^ and 10^−5^ of FC (shown as an example) and the dose–response curves for the FC (10^−9^−10^−5^ M)-induced total elongation growth of the coleoptile segments as a function of time (Figure 1B).

When fusicoccin (FC) was added to the control medium at 10^−6^ or 10^−5^ M, it induced rapid growth (with a maximal growth rate of about 0.19 µm s^−1^ cm^−1^ for both concentrations), the kinetics of which was parabolic shaped. At lower concentrations (10^−9^–10^−7^ M), the acceleration of rapid growth was lowered. Based on the growth rate responses that are shown in Figure 1A (to ensure the clarity of the figure, only some of the curves are shown), the total elongation growth over 10 h was calculated (Figure 1A, inset). Taking into account the results of all of the growth experiments that were performed in a wide range of FC concentrations (10^−9^–10^−5^ M), the dose–response curves for the FC-induced elongation growth as a function of time (elongation measured three and eight hours after the addition of FC) were constructed (Figure 1B). The data in Figure 1B indicate that when the maize coleoptile segments were incubated in the presence of the various FC concentrations, there were sigmoidal-shaped dose–response curves, regardless of the duration of the incubation of the segments in the presence of FC. However, as can be seen in Figure 1B, fusicoccin at concentrations of 10^−6^ and 10^−5^ M was maximal in the short- (3 h) and long-term (8 h) growth experiments (the values of the total elongation growth that were calculated for FC at 10^−6^ and 10^−5^ M were not significantly different, t-test). However, considering that at 10^−5^ M of FC there was a downward trend in growth over a longer period of time, FC at a concentration of 10^−6^ M should be considered to be optimal.

The data that was obtained for medium pH (Figure 2A), which was measured simultaneously with growth, indicated that the coleoptile segments that had been incubated in the control medium changed their pH in a manner that was dependent on the FC concentration and time.

Generally, within the first two to three hours, there was an increase of pH to 6.0–6.4, which in the control medium was followed by a slow decrease to a pH of approximately 5.0 after 10 h. When FC was added to the control medium, its effect on the medium pH depended on the FC concentration and time (Figure 2A). At the highest concentrations (10^−6^ and 10^−5^ M), FC caused a medium acidification below pH 4, while at the rest of the concentrations, the acidification of the medium was lower (Figure 2A). Based on the results of all of the pH experiments, the dose–response curves for the FC (10^−9^–10^−5^ M)-induced medium pH changes (expressed as the H^+^ concentration per coleoptile segment) as a function of time were constructed (Figure 2B). As can be seen in Figure 2B, FC at 10^−5^ M caused the maximal FC-induced proton extrusion per coleoptile segment. The regression constants and Pearson correlation coefficients (R = 0.9716 in the first time interval, Figure 2C, and R = 0.9998 in the second time interval, Figure 2D) that were calculated for FC (10^−6^)-induced elongation growth and proton extrusion confirmed the validity of the “acid growth theory” of fusicoccin action.

### 2.2. The Kinetics of FC- and IAA-Induced Elongation Growth, Medium pH and Membrane Potential Measured Simultaneously at Their Optimal and Suboptimal Concentrations

Figure 3A shows the comparison of the FC- and IAA-induced growth rate of the maize coleoptile segments recorded at their optimal and suboptimal concentrations. The data for auxin was adopted from our recently published paper [8]. As can be seen in Figure 3A, when FC was added at the optimal concentration of 10^−6^ M, it induced rapid growth with a maximal growth rate of 0.19 µm s^−1^ cm^−1^, which was observed about 60 min after the addition of FC. However, when FC was present at the suboptimal concentration of 10^−7^ M in the incubation medium, it induced growth with a maximal rate that was about 50% lower than the rate for the optimal FC concentration. In this case, the maximal growth rate was observed 120 min after the addition of FC. The total elongation growth of the maize coleoptile segments recorded at the optimal FC concentration was 30% higher than at the suboptimal concentration (Figure 3A, inset). However, when IAA was added to the incubation medium at the optimal or suboptimal concentration, the biphasic kinetics of the growth rate was observed (Figure 3A). IAA at the optimal concentration of 10^−4^ M induced strong growth with the maximum in the first and second phase equal 0.12 and 0.14 µm s^−1^ cm^−1^, respectively. IAA at the suboptimal concentration (10^−5^ M) induced growth with a maximum growth rate of about 0.15 µm s^−1^ cm^−1^ for both phases. The total elongation growth of the maize coleoptile segments that was calculated for the IAA that was added at the optimal concentration was 30% greater compared to its suboptimal concentration (Figure 3A, inset).

The data that was obtained for medium pH, which was measured simultaneously with growth (Figure 3B), indicated that the coleoptile segments that had been incubated in the presence of the optimal and suboptimal concentrations of either FC or IAA changed the medium pH in a manner that was dependent on the concentrations of both effectors and time. When FC was added to the control medium 2 h after the start of the experiment at 10^−6^ M (concentration optimal for fusicoccin-induced growth), a decrease in pH to ca. 3.8 was observed. However, FC added to the control medium at the suboptimal concentration (10^−7^ M) changed the medium pH in a way that over the first 360 min was similar to the change that was observed in the control medium and after this time the medium pH decreased to the level that was observed for IAA at 10^−4^ M IAA. In turn, the IAA at the optimal concentration of 10^–4^ M when added to the control medium caused a decrease in pH to approximately 4.5 over 8 h. However, when the IAA at the suboptimal concentration (10^−5^ M) was added to the control medium at 120 min, there was a decrease in pH to ca. 5.0 within the next 220 min (with kinetics similar to that observed at 10^−4^ M of IAA), although after this time, there was a recovery of medium pH to the value of the control medium. Figure 3B (inset) shows the medium pH changes of the coleoptile segments, which are expressed as changes in the H^+^ concentration per coleoptile segment. As can be seen from the inset in Figure 3B, the proton extrusion at 600 min for FC at 10^−6^ M was about fourfold greater than for the optimal concentration of IAA (10^−4^ M).

In simultaneous measurements of growth, medium pH and membrane potential of the parenchymal coleoptile cells, we have recently shown [8] that IAA when added to the control medium at the suboptimal concentration of 10^−5^ M caused an initial depolarization of the membrane potential, which was followed by the membrane hyperpolarization during which the membrane potential was 27 mV more negative than the original value (Figure 3C). However, when the coleoptile segments were treated with the optimal IAA concentration (10^−4^ M), the character of the IAA-induced membrane potential changes was generally similar to the one that was observed at 10^−5^ M IAA, but the duration of the membrane hyperpolarization was prolonged. However, when the FC was added to the medium at the optimal concentration of 10^−6^ M, it caused a rapid hyperpolarization of the membrane potential, which at 30 min was ca. 25 mV more negative than the original value and did not change significantly within the next 45 min. FC at the suboptimal concentration of 10^−7^ M caused membrane hyperpolarization that was somewhat slower and lower compared to FC at 10^−6^ M.

The parameters that are characteristic for the growth, medium pH and membrane potential are listed in Table 1. As is indicated from Table 1, the lag time for all three responses was significantly shorter for FC compared to IAA. Moreover, the maximal growth rate of the FC-induced rapid response was 60–70% greater than for IAA. The data in Table 1 indicates that the growth rate of the coleoptile segments that was calculated at the optimal concentration of FC (10^–6^ M) in the time interval of 300–420 min was 36 % lower than the one at the time interval of 150–270 min, while the rate of proton extrusion that was calculated at the time interval of 300–420 min was threefold higher compared to the time interval of 150–270 min. In the case of the optimal IAA concentration (10^−4^ M), similar relationships to those for FC were also found; the growth rate of the coleoptile segments that was observed at the time interval of 300–420 min was 13 % lower compared to the one at the time interval of 150–270 min, while the rate of proton extrusion at the time interval of 300–420 min was fivefold higher than that at the time interval of 150–270 min.

### 2.3. High Temperatures Change the Biphasic Kinetics of IAA-Induced Growth to a Monophasic One

In this section, the data that shows how temperature changes the kinetics of the growth rate response is described. Figure 4A shows that when IAA was added to the control medium at the optimal concentration of 10^−4^ M (2 h after the start of the experiment), it induced a growth response, in which the kinetics and total elongation growth of the coleoptile segments depended on the medium temperature in the range 15–30 °C and time.

At the lowest temperature (15 °C), the growth response of the coleoptile segments that had been treated with IAA had the character of a monotonically increasing function whose values did not exceed 0.06 µm s^−1^ cm^−1^. However, when the coleoptile segments were incubated at higher temperatures, 20 and 25 °C, the kinetics of the growth response could be divided into two phases (biphasic response); the first, which was very rapid, was followed by a long-lasting phase. At 20 °C, the first phase, which had a maximal growth rate of ca. 0.08 µm s^−1^ cm^−1^ (which took about 40 min), was followed by the second phase, which had a maximal growth rate of 0.11 µm s^−1^ cm^−1^ at 310 min. For the coleoptile segments that had been incubated at 25 °C, the maxima of both phases were higher and moved to the left compared to those at 20 °C. At the highest temperature (30 °C), the biphasic character of the growth response of the coleoptile segments that had been treated with IAA changed to a monophasic response whose shape was similar to the one that was induced by the optimal concentration of FC (Figure 4A). Interestingly, at 30 °C, the amplitude of the IAA-induced growth was significantly higher (by 26%) than for FC at 25 °C. As can be seen in Figure 4A (inset), the total elongation growth of the maize coleoptile segments increased with an increasing medium temperature. For example, the total elongation growth at 30 °C was threefold greater compared to that at 15 °C (Figure 4A, inset). Figure 4A also shows that the lag time for auxin-induced growth can be shortened by increasing the temperature and that at 30 °C, it was practically the same as for FC.

The data that was obtained for the medium pH changes (Figure 4B), which was measured simultaneously with growth, indicated that when IAA was added to the medium at 30 °C, there was a rapid acceleration of proton extrusion by the coleoptile segments whose steepness (expressed as the slope of the tangent to the proton extrusion curve from 150 to 240 min, Figure 4B) was 0.222 pM s^−1^ cm^−1^, while for the coleoptile segments that had been incubated at 25 °C the slope of the tangent was twofold lower (0.111 pM s^−1^ cm ^−1^). For comparison, the rate of proton extrusion that was found in the presence of the optimal concentration of FC at 25 °C was 1.172 pM s^−1^ cm^−1^.

## 3. Discussion

It is well established that the fungal toxin fusicoccin (FC) and the natural auxin indole-3-acetic acid (IAA) both induce rapid growth and proton extrusion in coleoptile cells [3,4,5]. It should also be added that coleoptile segments represent a classical model system for studies on the elongation growth of plant cells and that much of the information about the mechanism of auxin action was obtained using this model system. The main question that is addressed here is whether the ionic mechanisms of FC- and IAA-induced rapid growth differ or not.

The dose–response curve for the FC-induced elongation growth of the maize coleoptile segments that was obtained here was sigmoidal-shaped with the optimum at 10^−6^ M, regardless of the duration of the incubation of the segments in the presence of FC (Figure 1B). The symmetrical shift of the dose–response curve that was observed between 3 and 8 h might suggest that the same molecular mechanism of fusicoccin action is involved at these time intervals. A similar shape of the dose–response curves for the FC-induced growth of coleoptile segments was also reported earlier by others [29,30,31]. The dose–response curves for the FC-induced proton extrusion (expressed as H^+^ concentration per coleoptile segment) that were obtained here, which were measured simultaneously with the growth, were also sigmoidal-shaped, but in this case, the maximal response (rather than the optimal) was observed at 10^−5^ M FC (Figure 2B). This concentration differed by one order from the one that was found for the optimal value of FC-induced elongation growth. A similar phenomenon was also reported by Cleland [31], who suggested that the maximal rate of FC-induced proton extrusion is greater than is needed to saturate the growth system. For comparison, the dose–response curves, which were recently obtained by us [8] for IAA-induced growth and proton extrusion, were bell-shaped curves with the optimal concentration at 10^−4^ M IAA.

To determine the possible differences between the ionic mechanisms of FC- and IAA-induced elongation growth of the maize coleoptile segments, we quantitively compared three parameters, namely, elongation growth, medium pH and membrane potential, which were simultaneously measured on the same tissue sample. These parameters were compared at the optimal concentrations of both effectors (Table 1). FC and IAA at the concentrations that are optimal for growth, 10^–6^ and 10^–4^ M, respectively, induced practically the same total elongation growth of the maize coleoptile segments (ca. 2800–3000 µm cm^−1^ over 8 h after the addition of FC or IAA, see also [8]), while the medium pH (expressed as changes in the H^+^ concentration *per* coleoptile segment) (Figure 2B) was about fourfold greater for FC compared to IAA (Figure 2B). However, when the rate of proton extrusion in the initial phase of growth (expressed as pM s^−1^ cm^−1^ and calculated between 150 and 270 min) was taken into account, it was ninefold higher for FC than for IAA (Table 1). The question was whether the rate of elongation growth of coleoptile segments and the rate of proton extrusion that are determined in the presence of FC and IAA are correlated at selected time intervals (Table 1). As can be seen in Table 1, the growth rate of the coleoptile segments which was calculated at the optimal concentration of FC (10^–6^ M) in the time interval of 300–420 min, was 36 % lower than the one in the time interval of 150–270 min, while the rate of proton extrusion, which was calculated in the time interval of 300–420 min, was threefold higher compared to the one for 150–270 min. In the case of the optimal IAA concentration (10^−4^ M), similar relationships as for FC were also found; the growth rate of the coleoptile segments that were observed in the time interval of 300–420 min was 13% lower compared to one in the time interval of 150–270 min, while the rate of proton extrusion in the time interval of 300–420 min was fivefold higher than that in the one in 150–270 min. The observation that the rate of elongation growth decreased with the increasing time, while the rate of proton extrusion increased might suggest that factors other than proton extrusion are involved in decreasing the growth rate of coleoptile segments. Taking into account the model that was proposed by Becker and Hedrich [9], this decrease in growth rate is probably caused by a drop in *zmk1* mRNA (see Introduction).

When the FC was added to the control medium at the optimal concentration (10^–6^ M), it caused a rapid hyperpolarization of the membrane potential, which at 30 min was about 25 mV more negative than the original value and did not change significantly within the next 45 min. This finding is a good agreement with the results of others [15,32,33,34]. At present, there is no doubt that plasma membrane hyperpolarization in the presence of FC or IAA is caused by the stimulation of proton extrusion by H^+^-ATPase [33,35]. For comparison, when FC was added at the suboptimal concentration, it induced membrane hyperpolarization, which was significantly lower (about 5 mV) and slower (about 0.7 mV/min) compared to the effect of the optimal FC concentration. However, when IAA was added to the control medium at the optimal IAA concentration (10^−4^ M), it caused an initial depolarization of the membrane potential, which was followed by the transient membrane hyperpolarization (within 50 min) during which the membrane potential was 22 mV more negative than the original value. However, when the coleoptile segments were treated with the suboptimal IAA concentration (10^–5^ M), the kinetics of the auxin-induced membrane potential changes was generally similar to the one that was observed at the optimal IAA concentration (10^−4^ M) except that the duration of the membrane hyperpolarization was very short (5–6 min) (Figure 3C). The main differences between the membrane potential changes that were observed at the optimal concentrations of FC and IAA relate to the permanent membrane hyperpolarization in the presence of FC and the transient hyperpolarization in the presence of IAA. It has recently been shown that auxin-induced plasma membrane depolarization is regulated by auxin transport [36].

The higher amplitude of the FC-induced rapid growth can be explained by taking into account the recent findings that were published by Saporano et al. [26], who found that in addition to the well-studied effect of FC on PM H^+^-ATPase (reviewed in [37]), this fungal toxin also directly augments the activity of the K^+^ inward rectifier KAT1 (K^+^ *Arabidopsis thaliana* 1). In accordance with the hypothesis that was proposed by Saporano et al. [26], the binding of the 14−3−3 proteins to the MODE III binding site of KAT1 increases the conductance of the inward rectifier channel. In turn, the KAT1: 14−3−3 protein complex, to which the fungal toxin binds and stabilizes it, which causes K^+^ influx to be further augmented, is also a target of FC. The above means that fusicoccin has a common mechanism for regulating the proton pump and the potassium channel.

One unexpected finding of our experiments was determining that a high temperature (30 °C) changed the shape of the IAA-induced growth into one that is characteristic for FC-induced growth (Figure 4A). This phenomenon probably results from the two findings shown here, namely, the first, that at higher temperatures, the lag time of IAA-induced rapid growth is significantly shortened, and second, that the IAA-induced rapid growth and proton extrusion increased with increasing temperature. These two findings were also reported by others [29,38]. There is also a third possibility, namely, that higher temperatures increase the conductance of the hyperpolarization-activated K^+^ inward channels as a result of the negative shift (hyperpolarization) of the membrane potential. This third possibility was demonstrated by Ilan et al. [39] in the elegant patch-clamp experiments that were performed with *Vicia faba* guard-cell protoplasts. The authors showed that the negative shift of E_1/2_ (half activation voltage) activated the K^+^ uptake channels with an increase of the temperature between 20 and 28 °C. The hyperpolarization of the membrane potential at higher temperatures (30–40 °C) was also found in maize coleoptile cells [38].

## 4. Materials and Methods

### 4.1. Plant Material

Caryopses of maize (Zea mays L. cv. Koka, Kobierzyce, Poland) were soaked in tap water, sown on wet lignin in plastic boxes and placed in a growth chamber (Type MIR-553, Sanyo Electric Co., Osaka, Japan) at 27 ± 1.0 °C for four days in the dark. The experiments were performed with 10 mm long coleoptile segments that had been cut from maize seedlings (the length of each coleoptile was 2–3 cm) as was recently described by Polak and Karcz [8]. The coleoptile segments from which the first leaves had been removed were excised 3 mm below the tip and collected in an incubation medium containing (control medium) 1 mM KCl, 0.1 mM NaCl and 0.1 mM CaCl_2_. In all of the experiments, the initial pH of the control medium was adjusted to 5.8–6.0.

### 4.2. Chemicals

Indole-3-acetic acid (IAA) (Serva, Heidelberg, Germany) was used as potassium salt, since it could be rapidly dissolved in water. Fusicoccin (Sigma, St. Louis, MO, USA) was dissolved in ethanol and added to the incubation medium at needed concentration.

### 4.3. Growth, Medium pH and Membrane Potential Measurements

The experiments with the coleoptile segments were conducted in an apparatus that enabled their elongation growth, medium pH and membrane potential of the parenchymal cells from the same tissue sample to be simultaneously measured [8]. Briefly, in order to simultaneously measure the growth and medium pH, 60 coleoptile segments were arranged vertically in three narrow glass pipettes (20 segments in each); however, when the electrophysiological chamber was used, each pipette contained 22 segments (in order to increase the volume of the medium). The medium was circulated by a peristaltic pump (1B-05A; Zalimp, Warsaw, Poland). The high-resolution measurements of the growth rate were performed using an angular position transducer (TWK-Electronik, Düsseldorf, Germany). The coleoptile segments were incubated in an intensively aerated medium in which the volume of the incubation medium was constant (0.3 mL/segment). The incubation medium also flowed through the lumen of the coleoptile cylinders. This feature enabled the experimental solutions to be in direct contact with the interior of the segments, which significantly enhances both the elongation growth of the coleoptile segments and proton extrusion [8]. The extension growth of a stack of 20 (or 22) segments and the pH of the incubation medium were sampled every 3 min using a multifunctional computer meter (CX-771; Elmetron, Zabrze, Poland). The pH was measured with a pH electrode (OSH 10-10; Metron, Torun, Poland). All of the manipulations, growth and pH measurements were conducted under dim green light at a thermostatically controlled temperature.

### 4.4. Electrophysiological Experiments

The electrophysiological experiments were conducted on intact 10 mm long maize coleoptile segments that were prepared in the same manner as for the growth experiments. The experimental technique that was recently described by Polak and Karcz [8] was used. Briefly, the membrane potential (Em) was measured by recording the voltage between a 3 M KCl-filled glass micropipette inserted into the parenchymal cells and a reference electrode in the bathing medium. For the electrophysiological experiments, the segments were vertically placed into a perfusion Plexiglas chamber that was mounted on a microscope stage (for details, see [8]). The microelectrodes were inserted into the cells under the microscope using micromanipulator (Hugo Sachs Electronik, March-Hugstteten, Germany). The micropipettes were made from borosilicate glass capillaries (type 1B150F-3; World Precision Instruments, Sarasota, FL, USA) using a vertical pipette puller (Model PIP 6; HEKA Elektronik, Lambrecht, Germany).

### 4.5. Statistical Analysis

The data were analyzed using Statistica software for Windows (STATISTICA data analysis software system, version 13.1 http://www.statsoft.com, accssed on 8 May 2021, Tulsa, OK, USA). The Student’s t-test was used to evaluate the significance of the differences between the membrane potential values. The correlation of the elongation growth and proton concentrations was calculated based on Pearson’s correlation.

## 5. Conclusion

Maize coleoptile segments respond to the optimal concentrations of FC (10^−6^ M) and IAA (10^−4^ M) by an increase in the elongation growth to a similar level over 10 h, although both effectors differ in the kinetics of initial growth. Simultaneously with growth, the medium pH that was measured indicated that FC at 10^−6^ M induced proton extrusion, which was about fourfold greater at 10 h than for the optimal concentration of IAA. In the initial phase of growth, the rate of proton extrusion was ninefold higher for FC than for IAA. The electrophysiological experiments showed that FC at the optimal concentration caused permanent membrane hyperpolarization, while in the presence of the optimal IAA, the hyperpolarization was transient. It was also found that the lag time of the rapid growth, proton extrusion and membrane hyperpolarization were significantly shorter for FC compared to IAA. At 30 °C, the biphasic kinetics of the IAA-induced growth rate changed into a monophasic (parabolic) one, which is characteristic for FC-induced rapid growth. This suggests that the rates of the initial phase of the FC- and IAA-induced growth involve two common mechanisms that consist of the proton pumps and potassium channels whose contribution to the action of both effectors on the rapid growth is different.

## Figures and Tables

**Figure 1 ijms-22-05017-f001:**
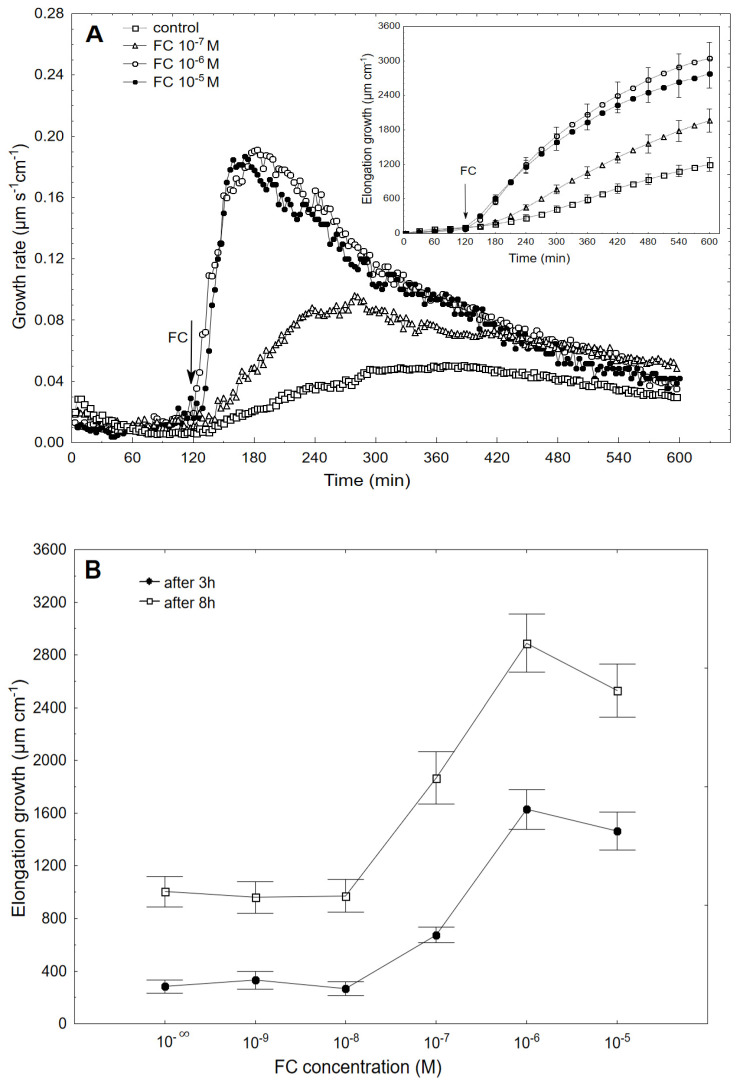
Growth rate (µm s^−1^cm^−1^) of the maize coleoptile segments that had been incubated in the presence of 10^−7^, 10^−6^ and 10^−5^ M of FC (**A**) and the dose–response curves for the FC (10^−9^–10^−5^ M)-induced total elongation growth of the coleoptile segments as a function of time (**B**). The coleoptile segments were first preincubated (over 2 h) in a control medium to which FC had been added (arrow). The inset in Figure 1A shows the total elongation growth (µm cm^−1^), which was calculated as the sum of the extensions from 3 min interval measurements over 10 h. All of the curves are the means of at least five independent experiments. Bars indicate ± SE.

**Figure 2 ijms-22-05017-f002:**
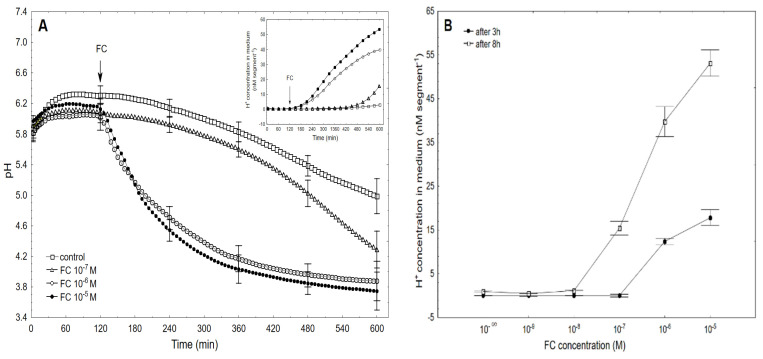
Kinetics of the medium pH changes of the coleoptile segments that had been incubated in the presence of 10^−7^, 10^−6^ and 10^−5^ M of FC (**A**), the dose–response curves for the FC (10^−9^–10^−5^ M)-induced medium pH changes (expressed as changes in the H+ concentration per coleoptile segment, nM segment^−1^ or nM cm^−1^) of the coleoptile segments as a function of time (**B**), and correlation between elongation (µm segment^−1^) and proton extrusion (nM segment^−1^) for two time intervals; the first, from the addition of FC to the end of the experiment (120–600 min) (0 < x < 40) **(C),** and the second starting 30 min later (150–600 min) (x > 9) **(D)**. In the second time interval, the FC-induced rapid growth was omitted. The inset in Figure 2A shows the proton extrusion, which was expressed as the changes in the H^+^ concentration per coleoptile segment. The coleoptile segments were first preincubated (over 2 h) in the control medium to which FC was added (arrow). The pH values are the means of at least five independent experiments that were performed simultaneously with growth (shown in Figure 1) using the same tissue samples. Bars indicate ± SE.

**Figure 3 ijms-22-05017-f003:**
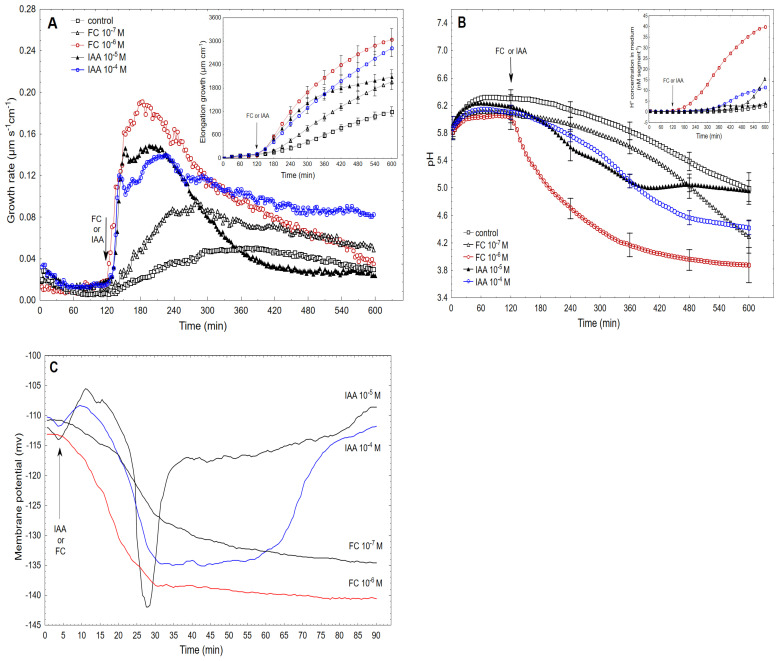
Kinetics of the FC- and IAA-induced growth rate (**A**), medium pH (**B**) and membrane potential (**C**) measured simultaneously at their optimal and suboptimal concentrations. The data for IAA-induced growth, proton extrusion and membrane potential changes was adopted from our recently published paper [8]. The growth rate (µm s^−1^ cm^−1^) of the maize coleoptile segments that had been incubated in the presence of the suboptimal (10^−7^ M) and optimal concentrations of FC (10^−6^ M, red curve) and both the suboptimal (10^−5^ M) and optimal (10^−4^ M, blue curve) concentrations of IAA are shown. The coleoptile segments were first preincubated (over 2 h) in the control medium to which FC or IAA was added (arrow). The insets show the total elongation growth (µm cm^−1^), which was calculated as the sum of the extensions from 3 min interval measurements over 10 h (Figure 3A) and the proton extrusion is expressed as the changes in the H^+^ concentration per coleoptile segment (Figure 3B). The medium pH changes were measured simultaneously with growth (Figure 3A) using the same tissue samples (red and blue curves as in Figure 3A). The membrane potential changes of the parenchymal coleoptile cells, which were simultaneously measured with the growth and medium pH changes and recorded at the suboptimal and optimal concentrations of FC and IAA are shown. All of the curves are the means of at least five independent experiments. Bars indicate ± SE.

**Figure 4 ijms-22-05017-f004:**
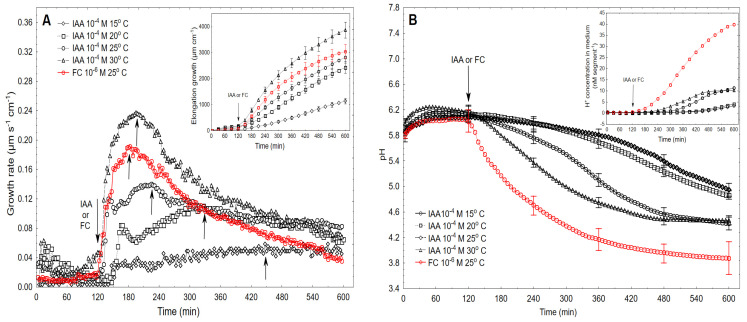
Effect of temperature (15, 20, 25 and 30 °C) on the growth rate (**A**) and medium pH (**B**) of the maize coleoptile segments that had been incubated in the presence of the optimal concentration of the IAA (10^−4^ M). For comparison, the effect of the optimal concentration of FC (10^−6^ M, red curve) at 25 °C is also shown. After preincubation (over 120 min at the desired temperature) of the coleoptile segments in the control medium, IAA was added (arrow). The arrow pointing up indicates movement to the left the second phase of the IAA-induced growth and the maximal growth rate that was induced by FC. The inset in Figure 4A shows the total elongation growth (µm cm^−1^), which was calculated as the sum of the extensions from 3 min interval measurements over 10 h; however, the one in Figure 4B shows the medium pH changes expressed as changes in the H^+^ concentration per coleoptile segment. The medium pH changes were measured simultaneously with growth (Figure 4A) using the same tissue samples. All of the curves are the means of at least seven independent experiments. Bars indicate ± SE.

**Table 1 ijms-22-05017-t001:** Parameters of the FC (10^−6^ M)- and IAA (10^−4^ M)-induced rapid growth, medium pH and membrane potential of the maize coleoptile cells that had been incubated at the optimal concentrations of both effectors at 25 °C.

Rapid growth, medium pH and membrane potential in the maize coleoptile cells that had been treated with the optimal concentrations of FC and IAA for growth					
Optimal Concentration of FC or IAA	Parameters under Consideration	Initial Value of Response	Lag Time of Response (min)	Maximal Value of Response	Rate of Response Approximated by its Steady State (from … to …min)
FC at 10^−6^ M	Growth rate µm s^−1^ cm^−1^	<0.04	3	0.19–0.20	0.166 ± 0.014 from 150 to 270 0.101 ± 0.007 from 300 to 420
	Medium pH [H^+^] cm^−1^ nM cm^−1^	0.3 ± 0.071 (*n* = 7, ±SE)	3	40 ± 3.7 (*n* = 7, ± SE) over 600 min	0.88 pM s^−1^ cm^−1^ from 150 to 270 2.51 pM s^−1^ cm^−1^ from 300 to 420
	Membrane potential mv	−114.1 ± 4.6 (*n* = 5, ± SE)	1–2	−139 ± 5.6 (*n* = 5, ± SE)	1.13 mV min^−1^ from 15 to 30
IAA at 10^−4^ M	Growth rate µm s^−1^ cm^−1^	<0.04	12–15	0.12	0.111 ± 0.012 from 150 to 270 0.097 ± 0.007 from 300 to 420
	Medium pH [H^+^] cm^−1^ nM cm^−1^	0.28 ± 0.084 (*n* = 9, ± SE)	20–30	11 ± 0.9 (*n* = 9, ± SE) over 600 min	0.096 pM s^−1^ cm^−1^ from 150 to 270 0.49 pM s^−1^ cm^−1^ from 300 to 420
	Membrane potential mv	−113.3 ± 4.1 (*n* = 8, ± SE)	3–4	−135±5.1 (*n* = 5, ± SE)	1.5 mV min^−1^ from 15 to 30

## Data Availability

The data that support the findings of this study are available from the corresponding author upon reasonable request.

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
