# Peer review of "Fusicoccin (FC)-Induced Rapid Growth, Proton Extrusion and Membrane Potential Changes in Maize (Zea mays L.) Coleoptile Cells: Comparison to Auxin Responses"

_ijms, 2021, doi:10.3390/ijms22095017_

Round 1
Reviewer 1 Report
The review on the publication by Polak, Karcz under the title ‘Fusicoccin (FC)-induced rapid growth, proton extrusion and membrane potential changes in maize (Zea mays L.) coleoptile cells: Comparison to auxin responses’
In my opinion the current work is very basic and not suitable for publishing in IJMS. The results of the experiments are not sufficient enough to compare both mechanism of action by FC and by IAA. To made such a deep conclusion authors have to add some experiments on the maize mutants that already are available in the literature (for example Richardson et al, 2020; Galli et al 2015).
Line 27 Auxin is not IAA. IAA - Indole-3-acetic acid is the most common naturally occurring plant hormone of the auxin class.
Line 391 Vicia faba please change to italics.
Author Response
Response to Reviewer 1
We sincerely thank the Reviewer 1 for constructive criticisms and valuable comments.
Reviewer 1 wrote:
The review on the publication by Polak, Karcz under the title ‘Fusicoccin (FC)-induced rapid growth, proton extrusion and membrane potential changes in maize (Zea mays L.) coleoptile cells: Comparison to auxin responses’
In my opinion the current work is very basic and not suitable for publishing in IJMS. The results of the experiments are not sufficient enough to compare both mechanism of action by FC and by IAA. To made such a deep conclusion authors have to add some experiments on the maize mutants that already are available in the literature (for example Richardson et al, 2020; Galli et al 2015).
Line 27 Auxin is not IAA. IAA - Indole-3-acetic acid is the most common naturally occurring plant hormone of the auxin class.
Line 391 Vicia faba please change to italics.
Responses and comments:
- Line 27 – we corrected this, now is: Auxin (IAA)
- Line 391 -we changed this, now is: Vicia faba
Our comments:
We agree with the reviewer's opinion that our work is very basic. But it is also true that the work deals with the basic problems concerning the ionic mechanisms of IAA- and FC-induced elongation growth of plant cells. The question is whether the ionic mechanism of action of both effectors on plant cell growth is the same or different. This question was also raised by the spiritual father of the acid growth theory, Robert Cleland, in the late seventies and eighties of the last century (Cleland, 1976; Rubinstein and Cleland, 1981). The answer to this question became partially possible after discoveries made at the turn of the century (described in the introduction to our work). However, the ionic mechanism of IAA and FC action on plant cell growth still raises a lot of controversy. For example, Kutschera's works question the validity of the acid growth theory in the case of IAA action and support its validity in the case of FC action (Kutschera and Schopfer, 1985a,b; Kutschera, 1994, 2006; Kutschera and Khanna, 2020). In this context our work shows the differences between the rapid phase of FC- and IAA-induced growth and suggests the reasons for the differences observed in the kinetics of growth response to IAA and FC. It appears that the key to solve these problems may partly lie in research on the contribution of ion transport across the vacuolar membrane of the plant cell and the impact of IAA and FC on this transport. Quite recently we have shown (Burdach et al., 2018, 2020) that the SV (slow vacuolar) and FV (fast vacuolar) channels, which represent the conductance of the major cations across the tonoplast, are involved in the IAA-induced volume changes of the vacuoles. We now starting the experiments on the influence of FC on the activity of the SV and FV channels.
In our present studies concerning the effect of FC on growth, proton extrusion and membrane potential in maize coleoptile cells we suggested that the rates of the initial phase of FC- and IAA-induced growth involve two common mechanisms that consist of the proton pumps and potassium channels whose contribution to the action of both effectors on the rapid growth is different.
Cleland, R. Fusicoccin-induced growth and hydrogen ion excretion of Avena coleoptiles: Relation to auxin responses. Planta 1976, 128, 201-206.
Rubinstein, B.; Cleland, R. Responses of Avena coleoptiles to suboptimal fusicoccin: Kinetics and comparisons with indoleacetic acid. Plant Physiol. 1981, 68, 543-547.
Rayle, D.L.; Cleland, R.E. The acid‐growth theory of auxin induced cell elongation is alive and well. Plant Physiol. 1992, 99, 1271–1274.
Kutschera, U.; Schopfer, P. Evidence against the acid‐growth theory of auxin action. Planta 1985a, 163, 483–493.
Kutschera, U.; Schopfer, P. Evidence for the acid‐growth theory of fusicoccin action. Planta 1985b, 163, 494–499.
Rayle, D.L.; Cleland, R.E. The acid‐growth theory of auxin induced cell elongation is alive and well. Plant Physiol. 1992, 99, 1271–1274.
Lüthen, H.; Bigdon, M.; Böttger, M. Re‐examination of the acid‐growth theory of auxin action. Plant Physiol. 1990, 93, 931–939.
Kutschera, U. The current status of the acid‐growth hypothesis. New Phytol. 1994, 126, 549–569.
Kutschera, U. Acid growth and plant development. Science 2006, 311, 952-953.
Kutschera, U.; Wang, Z-Y. Growth-limiting proteins in maize coleoptiles and the auxin-brassinosteroid hypothesis of mesocotyl elongation. Protoplasma 2015, 253, 3-14.
Kutschera, U.; Khanna, R. Auxin action in developing maize coleoptiles: challenges and open questions. Plant Signal. Behav. 2020, doi: 10.1080/15592324.2020.1762327
Burdach, Z.; Siemieniuk, A.; Trela, Z.; Kurtyka, R.; Karcz, W. Role of auxin (IAA) in the regulation of slow vacuolar (SV) channels and the volume of red beet taproot vacuoles. BMC Plant Biol. 2018, 18, doi: 10.1186/s12870-018-1321-6
Burdach, Z.; Siemieniuk, A.; Karcz, W. Effect of auxin (IAA) on the fast vacuolar (FV) channels in red beet (Beta vulgaris L.) taproot vacuoles. Inter. J. Mol. Sci. 2020, 21, doi:10.3390/ijms21144876
Reviewer 2 Report
The authors of the manuscript “Fusicoccin (FC)-induced rapid growth, proton extrusion and membrane potential changes in maize (Zea mays L.) coleoptile cells: Comparison to auxin responses” investigated the effects of FC and indole-3-acetic acid on the growth, medium pH and membrane potential of maize coleoptile cells. It is well written in general, and the topic is engaging.
Following are some suggestions for improvement of this manuscript:
Full names of the abbreviations can be given for the first time in the article writing (e.g. L14: IAA)
Please italicize all Latin terms throughout the text (e.g. L70: Fusicoccum amygdali).
L153-156: Data Pearson correlation coefficient is not provided.
L164: 0.19 μm s-1 cm-1 (Typo). Please correct it throughout the text.
L173: 10-4 (Typo). Please correct it throughout the text.
L291: Please correct the unit “(μm cm−1)”
Materials and methods
Please indicate the number of replicates used for statistical treatments in each experiment.
Please indicate the source of FC and introduce methods for IAA and FC treatment.
Reviewer 3 Report
Line 15: what is the "optimal concentration "of FC? Please, clarify that it is a maximal growth rate.
Line 39: why [9,5], not [5,9]? Please, close the bracket in this line.
Line 46: "newly synthesized K+ channels "– channels themselves can not synthesized. Please, re-formulate.
Line 68: "we compared the experiments "it is not very good sentence. I think you compare effects, not experiments.
Line 37: for the acid growth theory, you have to consider three simultaneously occurred events: cell wall loosening, plasma membrane loosening, and vacuolar membrane loosening. This can happen only in a cell which lytic vacuole.
I am not sure the word "optimal "is suitable here. I would suggest to use another term.
Figure 1, panel B: please, made a better layout to clarly show 3 and 8 hours.
Lines 98-99: curves is a figure description, for the results subsection it is better to give another title.
Line 100: what is: "High-resolution measurements of the medium pH "? Please, separate pH and growth.
Luie 115: "the acceleration of rapid growth was decreased "? Please, describe better.
Line 125: I am not sure "optimal ", it rather maximal.
Figure2: can you explain why in the control medium pH was higher before addition of FC?
Line 156: From the acid growth theory, it will also be essential to measure vacuolar pH and vacuolar membrane status: "vacuolar growth "accompanied by vacuole acidification and vacuolar membrane loosening.
Line 173: layout for 10-4 M.
Actually, 100 µM of IAA is not a physiological concentration, it may be a side effect of auxin. Auxin response pathway may activate at 1000 times fewer concentrations. I would rather tell about the effects of high IAA application, but not as auxin.
Line 198: I am not sure you need to repeat always this point: "concentration optimal for fusicoccin-induced growth ".
Line 254: layout 10-4 M
Line 301: I am not sure about "auxin action ": auxin responses have been detected at 1000 less concentration. For the correct conclusion, it will be important to use also 1-NAA 8as active auxin) and 2.NAA as a molecule that does not have an auxin effect. This allows you to separate chemical and biological effects.
Table 1: it is unclear units you have used for pH. Which cm? Which unit?
Author Response
Response to Reviewer 3
We sincerely thank the Reviewer 3 for constructive criticisms and valuable comments, which were of great help in revising the manuscript. Accordingly, the revised manuscript has been systematically improved.
Reviewer 3 wrote:
Początek formularza
Line 15: what is the "optimal concentration "of FC? Please, clarify that it is a maximal growth rate.
Line 39: why [9,5], not [5,9]? Please, close the bracket in this line.
Line 46: "newly synthesized K+ channels "– channels themselves can not synthesized. Please, re-formulate.
Line 68: "we compared the experiments "it is not very good sentence. I think you compare effects, not experiments.
Line 37: for the acid growth theory, you have to consider three simultaneously occurred events: cell wall loosening, plasma membrane loosening, and vacuolar membrane loosening. This can happen only in a cell which lytic vacuole.
I am not sure the word "optimal "is suitable here. I would suggest to use another term.
Figure 1, panel B: please, made a better layout to clarly show 3 and 8 hours.
Lines 98-99: curves is a figure description, for the results subsection it is better to give another title.
Line 100: what is: "High-resolution measurements of the medium pH "? Please, separate pH and growth.
Luie 115: "the acceleration of rapid growth was decreased "? Please, describe better.
Line 125: I am not sure "optimal ", it rather maximal.
Figure2: can you explain why in the control medium pH was higher before addition of FC?
Line 156: From the acid growth theory, it will also be essential to measure vacuolar pH and vacuolar membrane status: "vacuolar growth "accompanied by vacuole acidification and vacuolar membrane loosening.
Line 173: layout for 10-4 M.
Actually, 100 µM of IAA is not a physiological concentration, it may be a side effect of auxin. Auxin response pathway may activate at 1000 times fewer concentrations. I would rather tell about the effects of high IAA application, but not as auxin.
Line 198: I am not sure you need to repeat always this point: "concentration optimal for fusicoccin-induced growth ".
Line 254: layout 10-4 M
Line 301: I am not sure about "auxin action ": auxin responses have been detected at 1000 less concentration. For the correct conclusion, it will be important to use also 1-NAA 8as active auxin) and 2.NAA as a molecule that does not have an auxin effect. This allows you to separate chemical and biological effects.
Table 1: it is unclear units you have used for pH. Which cm? Which unit?
Dół formularza
Responses and comments:
- Line 15 – we changed this sentence;
was: The differences between the membrane potential changes that were observed at the optimal concentration of FC and IAA relate to the permanent membrane hyperpolarisation in the presence of FC and transient hyperpolarisation in the presence of IAA.
now is: The differences between the membrane potential changes that were observed in the presence of FC and IAA relate to the permanent membrane hyperpolarisation for the FC and transient hyperpolarisation for the IAA.
- Line 39 – we improved this, now is [5,9]
- Line 46 – we reformulated the sentence;
was: In turn, this leads to the incorporation of the newly synthesised K+ channels into the plasma membrane and to an increase in the number of active K+ channels in the plasma membrane [10].
now is: In turn, this leads to an increase in the number of active K+ channels in the plasma membrane [10].
- Line 68 – we improved phrase “we compared the experiments” , now is : we compared the results of the experiments
- Line 87 – Reviewer 3 wrote:
“ for the acid growth theory, you have to consider three simultaneously occurred events: cell wall loosening , plasma membrane loosening, and vacuolar membrane loosening. This can happen only in a cell with lytic vacuole.
Our comments:
We agree with Reviewer 2 that for the acid growth theory applied to the lytic vacuoles three simultaneously occurred events: cell wall loosening, plasma membrane loosening and vacuolar membrane loosening should be considered. However, the historical and current data on acid growth leaves us with a model which is at once parsimonious yet provides new questions. At the core of the model is a decrease in apoplastic pH, which leads to changes in the cell wall, resulting in growth. Modern reformulation are extending the original acid growth theory by the inclusion of other factors. Dünser and Kleine-Vehn (2015) proposed a mechanism, which they baptized “the acid growth balloon theory”, whereby auxin-driven changes in vacuolar volume are the key player behind cell elongation, underlining the importance of ion transport in acid growth. According to the “acid growth balloon theory”, the growth of plant cell is the interplay between the intracellular space-filling “vacuolar balloon” and the required extracellular cell wall acidification/loosening. In this context, we have recently shown (Burdach et al., 2018, 2020) that the SV (slow vacuolar) and FV (fast vacuolar) channels, which represent the conductance of the major cations across the tonoplast, are involved in the IAA-induced volume changes of the vacuoles. In the literature there is no model describing the relationships between the molecular mechanisms of IAA-induced volume changes of the cell and the vacuole.
Dünser, K., Kleine-Vehn, J. Differential growth regulation in plants-the acid growth balloon theory. Current Opinion in Plant Biology, 2015, 28, 55-59.
Burdach, Z.; Siemieniuk, A.; Trela, Z.; Kurtyka, R.; Karcz, W. Role of auxin (IAA) in the regulation of slow vacuolar (SV) channels and the volume of red beet taproot vacuoles. BMC Plant Biol. 2018, 18, doi: 10.1186/s12870-018-1321-6
Burdach, Z.; Siemieniuk, A.; Karcz, W. Effect of auxin (IAA) on the fast vacuolar (FV) channels in red beet (Beta vulgaris L.) taproot vacuoles. Inter. J. Mol. Sci. 2020, 21, doi: 10.3390/ijms21144876
- Figure 1, panel B: - we changed the layout to clearly show 3 and 8 hours.
- Lines 98-99 – we changed the title for the results subsection,
now is: FC-induced elongation growth of maize coleoptile segments and medium pH changes measured simultaneously with the growth
- Line 100 – we separated pH and growth:
was: High-resolution measurements of the growth and medium pH were performed using an apparatus that was recently described by Polak and Karcz [8].
now is: High-resolution measurements of the growth were performed using an apparatus that was recently described by Polak and Karcz [8]. Simultaneously with growth medium pH changes were also measured.
- Line 115 – we changed the phrase “the acceleration of rapid growth was decreased”
now is: “the acceleration of rapid growth was lowered”
- Line 125 – we changed “optimal” to maximal
- Figure 2: can you explain why in the control medium pH was higher before addition of FC?
Our comment:
It is well established that the coleoptile segments characteristically change the external medium pH. This characteristic change of the external medium pH consisted of an initial increase of pH, that usually ended within 2 h at a pH near neutral (so-called “neutral peak”, Peters and Felle, 1991a), and a slow pH decrease to the level ca. 5.0 after 10 h. In our experiments within the first two hours there was an increase of pH to 6.0-6.4 (Figure 2A). It has also been suggested that medium pH experiments, lacking such characteristic values of pH changes, are indicative of poor experimental conditions (Peters et al., 1998). FC added to the incubation medium at “neutral peak” (after 2 hours), during the next 8 hours, decrease of pH to the value of ca. 3.8. Taking into account the definition of pH and pH changes in the range 6.0-6.4, the differences in [H+] in this pH range are very small.
The characteristic changes of the external medium pH were observed by others and also by us:
Peters WS, Felle H. 1991a. Control of apoplast pH in corn coleoptile segments. I. The endogenous regulation of cell wall pH. Journal of Plant Physiology137, 655–661.
Peters WS, Felle H. 1991b. Control of apoplast pH in corn coleoptile segments. II. The effect of various auxins and auxin analogues. Journal of Plant Physiology137, 691–696.
Peters WS, Lüthen H, Böttger M, Felle H. 1998. The temporal correlation of changes in apoplast pH and growth rate in maize coleoptile segments. Australian Journal of Plant Physiology 25, 31–35.
Karcz, W.; Burdach, Z. 2002. A comparison of the effects of IAA and 4-Cl-IAA on growth, proton secretion and membrane potential in maize coleoptile segments. Journal of Experimental Botany 53, 1089–1098.
Burdach, Z.; Kurtyka, R.; Siemieniuk, A.; Karcz, W. 2014. Role of chloride ions in the promotion of auxin-induced growth of maize coleoptile segments. Annals of Botany 114, 1023-1034.
Kurtyka, R.; Burdach, Z.; Siemieniuk, A.; Karcz, W. 2018. Single and combined effects of Cd and Pb on the growth, medium pH, membrane potential and metal content in maize (Zea mays L.) coleoptile segments. Ecotoxicology and Environmental Safety 161, 8–16.
Rudnicka, M.; Ludynia, M.; Karcz, W. 2019. The effect of naphthazarin on the growth electrogenicity, oxidative stress, and microtubule array in Z. mays coleoptile cells treated with IAA. Front. Plant Sci. doi:10.3389/fpls.2018.01940
- Line 156 – Reviewer wrote:
From the acid growth theory, it will also be essential to measure vacuolar pH and vacuolar membrane status: “vacuolar growth” accompanied by vacuolar acidification and vacuolar membrane loosening.
Our comment:
We agree with the opinion of Reviewer that above mentioned facts are essential for the “total acid growth theory”, the theory that will connect the phenomena occurring in the vacuole and the whole cell. There is no such theory in the literature, and we think that our recent works go in this direction.
Burdach, Z.; Siemieniuk, A.; Trela, Z.; Kurtyka, R.; Karcz, W. Role of auxin (IAA) in the regulation of slow vacuolar (SV) channels and the volume of red beet taproot vacuoles. BMC Plant Biol. 2018, 18, doi: 10.1186/s12870-018-1321-6
Burdach, Z.; Siemieniuk, A.; Karcz, W. Effect of auxin (IAA) on the fast vacuolar (FV) channels in red beet (Beta vulgaris L.) taproot vacuoles. Inter. J. Mol. Sci. 2020, 21, doi: 10.3390/ijms21144876
- Line – 173 – we corrected 10-4
- Reviewer also wrote:
Actually, 100 µM is not a physiological concentration, it may be a side effect of auxin . Auxin response pathway may be activate at 1000 times fewer concentrations. I would rather tell about the effects of high IAA application, but not as auxin.
Our comment:
We agree with the Reviewer`s opinion that 100 µM of the IAA is very high concentration, but it depends on a parameter called "coleoptile density" (introduced by Peters et al.,1991a), which determines the volume of the incubation solution per coleoptyle segment. In our experiments this parameter is equal 0.3 ml/segment (0.3 ml/cm), and as was showed earlier (Polak, 2010) and recently (Polak and Karcz, 2021) the concentration 100 µM of IAA is optimal for the elongation growth of the maize coleoptile segments, which was measured over 8 h in in our elongation- and pH-measuring apparatus (Polak and Karcz, 2021). For comparison, IAA at 10 µM is optimal at the same experimental conditions, however, in short-term (3 h) recordings (Karcz et al., 1990). It should be added that IAA at 10 µM is also optimal in long-term (8 h) experiments if the “coleoptile density” is equal 5 ml/segment (Polak et al., 2011). If coleoptile segments are packed densely enough, the reliable estimations of cell wall pH are possible. In this case higher concentrations of IAA should be applied. However, physiological auxin activities measured in bioassays might be controlled by inhibitor levels rather than auxin concentrations (Bruinsma and Hasegawa, 1990) and was also showed that exogenous IAA was rapidly down regulated by IAA-conjugate synthesis (Cohen and Bandurski, 1982). At least in maize coleoptile segments, substantial IAA degradation occurs when the ratio of fresh weight per volume of medium is high, as it needs to be in physiological tests of auxin effects on cell wall pH regulations. This fact obvious bears on the interpretation of dose-response curves. In addition, maize coleoptile cells posses effective IAA uptake mechanisms (Hertel, 1987). The net proton uptake in the presence of IAA at 10 µM IAA (recovery of medium pH), which was previously observed with maize coleoptile segments by others (Kutschera and Schopfer, 1985a, b; Peters and Felle, 1991a, b) and also by us (Polak, 2010; Polak and Karcz, 2021), is due to rapid IAA metabolism (Peters et al., 1997). Using higher IAA concentration (100 µM) this recovery disappeared and the IAA(100 µM)-induced elongation growth of the coleoptile segments was about 40% higher than for IAA at 10 µM. IAA at 100 µM was also used by us in other works (Rudnicka et al. 2018, Plant Growth Regulation; Rudnicka et al., 2019. Int. J. Mol. Sci.; Rudnicka et al., 2019. Frontiers in Plant Science).
Cohen, JD.; Bandurski, RS. 1982. Chemistry and physiology of the bound auxins. Annual Review of Plant Physiology, 33, 403-430.
Kutschera U, Schopfer P. 1985a. Evidence against the acid‐growth theory of auxin action. Planta 163, 483–493.
Kutschera U, Schopfer P. 1985b. Evidence for the acid‐growth theory of fusiccocin action. Planta 163, 494–499.
Hertel, R. 1987. Auxin transport: binding of auxins and phytotropins to the carriers. accumulation into and efflux from membrane vesicles. Plant hormone receptors, Springer
Bruinsma, J.; Hasegawa, K. 1990. A new theory of phototropism–its regulation by a light‐induced gradient of auxin‐inhibiting substances. Physiologia plantarum, 79,700-704.
Karcz W, Stolarek J, Pietruszka M, Malkowski E. 1990. The dose–response curves for IAA‐induced elongation growth and acidification of the incubation medium of Zea mays L. coleoptile segments. Physiologia Plantarum 80, 257–261.
Peters WS, Felle H. 1991a. Control of apoplast pH in corn coleoptile segments. I. The endogenous regulation of cell wall pH. Journal of Plant Physiology137, 655–661.
Peters WS, Felle H. 1991b. Control of apoplast pH in corn coleoptile segments. II. The effect of various auxins and auxin analogues. Journal of Plant Physiology137, 691–696.
Peters WS, Lommel C, Felle H. 1997. IAA breakdown and its effect on auxin‐induced cell wall acidification in maize coleoptile segments. Physiologia Plantarum 100,415–422.
Polak M. 2010. The interdependences between growth, medium pH and membrane potential in maize coleoptile segments incubated in the presence of auxin (IAA), fusicoccin (FC) and allicin. Katowice, Poland, University of Silesia, Doctoral thesis.
Polak M., Tukaj Z., Karcz, W. 2011. Effect of temperature on the dose-response curves for auxin-induced elongation growth in maize coleoptile segments. Acta Physiologiae Plantarum 33, 437-442.
- Line 198 – Reviewer wrote:
“I am not sure you need to repeat always this point: “concentration optimal for fusicoccin-induced growth”.
- We corrected this
Line 254 – we corrected this
Line 301 - Reviewer wrote:
“I am not sure about “auxin action”: auxin responses have been detected at 1000 less concentration. For the correct conclusion, it will be important to use also 1-NAA as active auxin and 2-NAA as a molecule that does not have an auxin effect. This allow you to separate chemical and biological effects”
Our comment:
We think that the answer to this question is contained in the answer to the previous question concerning 100 µM IAA.
Table 1. - Reviewer wrote:
It is unclear units you have used for pH. Which cm? Which unit?
Our answer:
[H+] cm-1 – concentration of H+ in the incubation medium per coleoptile segment (length 1 cm)
nM cm-1 – nanoMol, pM cm-1 - pikoMol
pM s-1cm-1 - rate of proton extrusion per coleoptile segment , we used these units earlier.
Round 2
Reviewer 1 Report
The manuscript, in my opinion is not suitable for the publication in IJMS. It is a basic manuscript, not enough results to demonstrate the mechanism of the fusicoccin action and it is descriptive.Author Response
Thank you once more for preparing the opinion.
Reviewer 3 Report
Thank you very much for great improvement of the text.
Now it is almost good and can be accepted.
Small points still need to be considered in the next investigations.
For example, comparison of inactive auxin analogue 2-NAA with active one 1-NAA.
It will be great also to include vacuoles pH like it was done here:
http://www.plantphysiol.org/content/129/4/1807.short. Maybe combination of different fluorescent dyes will be useful (rhd123 for membrane and BCECF for pH).
For the growth, it will be also essential to use growth (as volume), not elongation. Current methods allow to do this.
Author Response
We would like to thank you again for constructive criticisms and valuable comments, which will be taken into account in the future. In the next experiments, we will compare the effect of 2-NAA with 1-NAA and also include vacuolar pH.